# Learning Compositional Neural Programs with Recursive Tree Search and Planning

**Thomas Pierrot**
InstaDeep
t.pierrot@instadeep.com

**Guillaume Ligner**
InstaDeep
g.ligner@instadeep.com

**Scott Reed**
DeepMind
reedscot@google.com

**Olivier Sigaud**
Sorbonne Université
olivier.sigaud@upmc.fr

**Nicolas Perrin**
CNRS, Sorbonne Université
perrin@isir.upmc.fr

**Alexandre Laterre**
InstaDeep
a.laterre@instadeep.com

**David Kas**
InstaDeep
d.kas@instadeep.com

**Karim Beguir**
InstaDeep
kb@instadeep.com

**Nando de Freitas**
DeepMind
nandodefreitas@google.com

## Abstract

We propose a novel reinforcement learning algorithm, AlphaNPI, that incorporates the strengths of Neural Programmer-Interpreters (NPI) and AlphaZero. NPI contributes structural biases in the form of modularity, hierarchy and recursion, which are helpful to reduce sample complexity, improve generalization and increase interpretability. AlphaZero contributes powerful neural network guided search algorithms, which we augment with recursion. AlphaNPI only assumes a hierarchical program specification with sparse rewards: 1 when the program execution satisfies the specification, and 0 otherwise. This specification enables us to overcome the need for strong supervision in the form of execution traces and consequently train NPI models effectively with reinforcement learning. The experiments show that AlphaNPI can sort as well as previous strongly supervised NPI variants. The AlphaNPI agent is also trained on a Tower of Hanoi puzzle with two disks and is shown to generalize to puzzles with an arbitrary number of disks. The experiments also show that when deploying our neural network policies, it is advantageous to do planning with guided Monte Carlo tree search.

## 1  Introduction

Learning a wide variety of skills, which can be reused and repurposed to learn more complex skills or to solve new problems, is one of the central challenges of artificial intelligence (AI). As argued in Bengio et al. [2019], beyond achieving good generalization when both the training and test data come from the same distribution, we want knowledge acquired in one setting to transfer to other settings with different but possibly related distributions.

Modularity is a powerful inductive bias for achieving this goal with neural networks [Parascandolo et al., 2018, Bengio et al., 2019]. Here, we focus on a particular modular representation known as Neural Programmer-Interpreters (NPI) [Reed and de Freitas, 2016]. The NPI architecture consists of a library of learned program embeddings that can be recomposed to solve different tasks, a core recurrent neural network that learns to interpret arbitrary programs, and domain-specific encoders for different environments. NPI achieves impressive multi-task results, with strong improvements in

generalization and reductions in sample complexity. While fixing the interpreter module, Reed and de Freitas [2016] also showed that NPI can learn new programs by re-using existing ones.

The NPI architecture can also learn recursive programs. In particular, Cai et al. [2017] demonstrates that it is possible to take advantage of recursion to obtain theoretical guarantees on the generalization behaviour of recursive NPIs. Recursive NPIs are thus amenable to verification and easy interpretation.

The NPI approach at first appears to be very general because as noted in [Reed and de Freitas, 2016], programs appear in many guises in AI; for example, as image transformations, as structured control policies, as classical algorithms, and as symbolic relations. However, NPI suffers from one important limitation: It requires supervised training from execution traces. This is a much stronger demand for supervision than input-output pairs. Thus the practical interest has been limited.

Some works have attempted to relax this strong supervision assumption. Li et al. [2017] and Fox et al. [2018] train variations of NPI using mostly low-level demonstration trajectories but still require a few full execution traces. Indeed, Fox et al. [2018] states "Our results suggest that adding weakly supervised demonstrations to the training set can improve performance at the task, but only when the strongly supervised demonstrations already get decent performance".

Xiao et al. [2018] incorporate combinatorial abstraction techniques from functional programming into NPI. They report no difficulties when learning using strong supervision, but substantial difficulties when attempting to learn NPI models with curricula and REINFORCE. In fact, this policy gradient reinforcement learning (RL) algorithm fails to learn simple recursive NPIs, attesting to the difficulty of applying RL to learn NPI models.

This paper demonstrates how to train NPI models effectively with RL for the first time. We remove the need for execution traces in exchange for a specification of programs and associated correctness tests on whether each program has completed successfully. This allows us to train the agent by telling it *what* needs to be done, instead of *how* it should be done. In other words, we show it is possible to overcome the need for strong supervision by replacing execution traces with a library of programs we want to learn and corresponding tests that assess whether a program has executed correctly.

The user specifying to the agent what to do, and not how to do it is reminiscent of programmable agents Denil et al. [2017] and declarative vs imperative programming. In our case, the user may also define a hierarchy in the program specification indicating which programs can be called by another.

The RL problem at-hand has a combinatorial nature, making it exceptionally hard to solve. Fortunately, we have witnessed significant progress in this area with the recent success of AlphaZero [Silver et al., 2017] in the game of Go. In the single-agent setting, Laterre et al. [2018] have demonstrated the power of AlphaZero when solving combinatorial bin packing problems.

In this work, we reformulate the original NPI as an actor-critic network and endow the search process of AlphaZero with the ability to handle hierarchy and recursion. These modifications, in addition to other more subtle changes detailed in the paper and appendices, enable us to construct a powerful RL agent, named AlphaNPI, that is able to train NPI models by RL[1].

AlphaNPI is shown to match the performance of strongly supervised versions of NPI in the experiments. The experiments also shed light on the issue of deploying neural network RL policies. Specifically, we find that agents that harness Monte Carlo tree search (MCTS) planning at test time are more effective than plain neural network policies.

## 2   Problem statement and definitions

We consider an agent interacting with an environment, choosing actions $a$ and making observations $e$. An example of this is bubble sort, where the environment is represented as a list of numbers, and the initial actions are one-step pointer moves and element swaps. We call this initial set of actions *atomic actions*. As training progresses, the agent learns to profit from atomic actions to acquire higher-level programs. Once a program is learned, it is incorporated into the set of available actions. For example, in bubble sort, the agent may learn the program RESET, which moves all pointers to the beginning of the list, and subsequently the agent may harness the program RESET as an action.

In our approach, a program has *pre-conditions* and *post-conditions*, which are tests on the environment state. All pre-conditions must be satisfied before execution. A program executes correctly if its post-conditions are verified. For example, the pre-condition for bubble sort is that both pointers are positioned at the beginning of the list. The post-condition is a test indicating whether the list is sorted upon termination. A program terminates when it calls the atomic action STOP, which is assumed to be available in all environments. A *level* is associated with each program, enabling us to define a hierarchy: Atomic actions have level 0 and any other program has a positive level. In our work, a program can only call lower-level programs or itself.

We formulate learning a hierarchical library of programs as a multi-task RL problem. In this setting, each task corresponds to learning a single program. The action space consists of atomic actions and learned programs. The reward signal is $1$ if a program executes correctly, and $0$ otherwise. The agent's goal is to maximize its expected reward over all the tasks. In other words, it has to learn all the programs in the input specification.

## 3 AlphaNPI

Our proposed agent, AlphaNPI, augments the NPI architecture of Reed and de Freitas [2016] to construct a recursive compositional neural network policy and a value function estimator, as illustrated in Figure 1. It also extends the MCTS procedure of Silver et al. [2017] to enable recursion. The

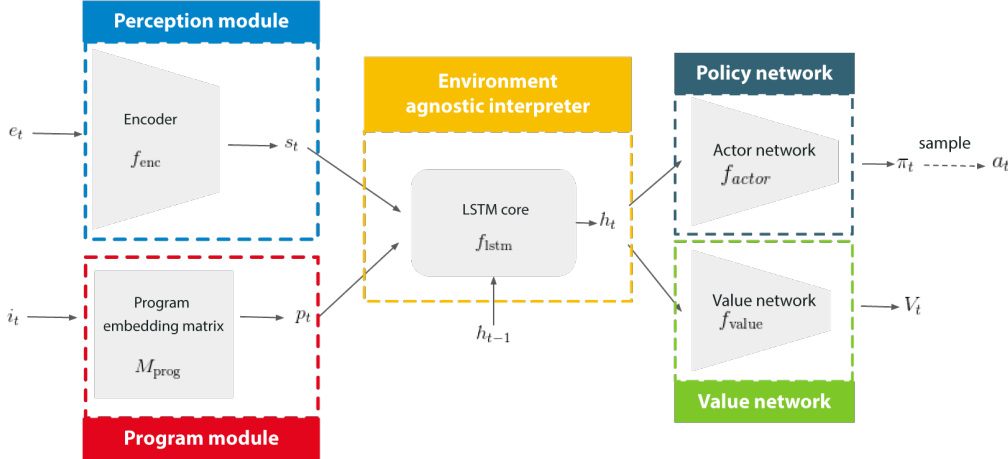

Figure 1: AlphaNPI modular neural network architecture.

AlphaNPI network architecture consists of five modules: State (or observation) encoders, a program embedding matrix, an LSTM [Hochreiter and Schmidhuber, 1997] interpreter, a policy (actor) network and a value network. Some of these modules are universal, and some are task-dependent. The architecture is consequently a natural fit for multi-task learning.

Programs are represented as vector embeddings $p$ indexed by $i$ in a library. As usual, we use an embedding matrix for this ($M_{\text{prog}}$). The observation encoder produces a vector of features $s$. The universal LSTM core interprets and executes arbitrary programs while conditioning on these features and its internal memory state $h$. The vector $p$ corresponding to index $i$ (stored in $M_{\text{prog}}$) is used by the LSTM core to know which program is being executed. A one-hot encoding of i could have been used instead, but the vector representation is more compact, and furthermore, since the components of $p$ are parameters of the network updated during training, their optimization can lead to generalization properties, as intuitively two programs with similar vector embeddings would yield relatively similar action decisions. The policy network converts the LSTM output to a vector of probabilities $\pi$ over the action space, while the policy network uses this output to estimate the value function $V$. The architecture is summarized by the following equations:

$$s_t = f_{\text{enc}}(e_t), \ p_t = M_{\text{prog}}[i_t, :], \ h_t = f_{\text{lstm}}(s_t, p_t, h_{t-1}), \ \pi_t = f_{\text{actor}}(h_t), \ V_t = f_{\text{value}}(h_t). \quad (1)$$

The neural nets have parameters, but we omit them in our notation to simplify the presentation. These parameters and the program embeddings are learned simultaneously during training by RL.

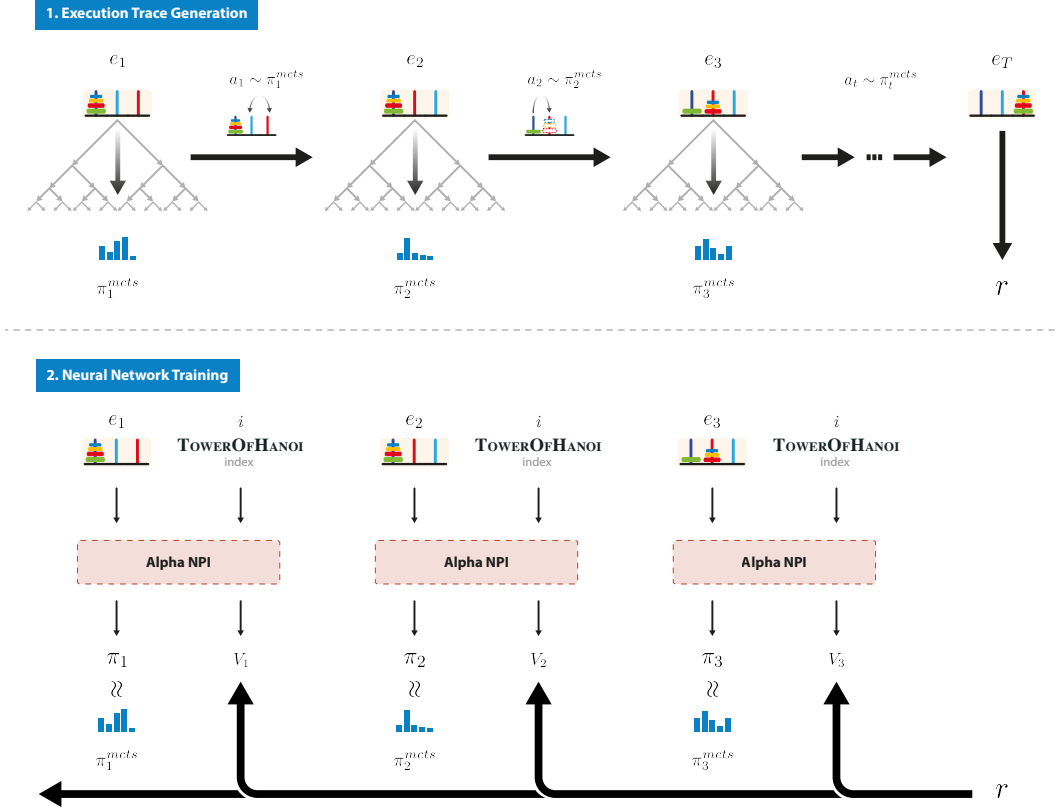

Figure 2: **Execution trace generation with AlphaNPI to solve the Tower of Hanoi puzzle. 1**. To execute the $i$-th program, TOWEROFHANOI, AlphaNPI generates an execution trace $(a_1, \ldots, a_T)$, with observations $(e_1, \ldots, e_T)$ produced by the environment and actions $a_t \sim \pi_t^{mcts}$ produced by MCTS using the latest neural net, see Figure 3. When the action STOP is chosen, the program's post-conditions are evaluated to compute the final reward $r$. The tuples $(e_t, i, h_t, \pi_t^{mcts}, r)$ are stored in a replay buffer. **2**. The neural network parameters are updated to maximise the similarity of its policy vector output $\pi$ to the search probabilities $\pi^{mcts}$, and to minimise the error between the predicted value $V$ and the final reward $r$. To train the neural network, shown in Figure 1, we use the data in the replay buffer.

When this AlphaNPI network executes a program, it can either call a learned sub-program or itself (recursively), or perform an atomic action. When the atomic action is STOP, the program terminates and control is returned to the calling program using a stack. When a sub-program is called, the stack depth increases and the LSTM memory state $h$ is set to a vector of zeroes. This turns out to be very important for verifying the model [Cai et al., 2017].

To generate data to train the AlphaNPI network by RL, we introduce a variant of AlphaZero using recursive MCTS. The general training procedure is illustrated in Figure 2, which is inspired by Figure 1 of Silver et al. [2017], but for a single-agent with hierarchical structure in this case. The Monte Carlo tree search (MCTS) guided by the AlphaNPI network enables the agent to "imagine" likely future scenarios and hence output an improved policy $\pi^{mcts}$, from which the next action is chosen[2]. This is repeated throughout the episode until the agent outputs the termination command STOP. If the program's post-conditions are satisfied, the agent obtains a final reward of 1, and 0 otherwise.

The data generated during these episodes is in turn used to retrain the AlphaNPI network. In particular, we record the sequence of observations, tree policies, LSTM internal states and rewards. We store the

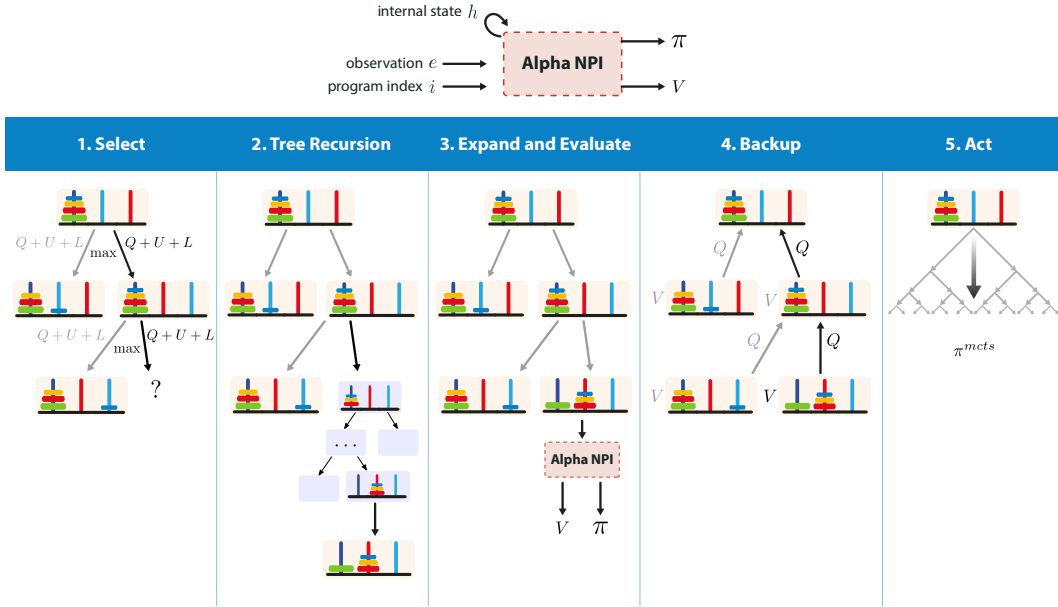

Figure 3: **Monte-Carlo tree search with AlphaNPI for the Tower of Hanoi puzzle. 1.** Each simulation traverses the tree by finding the actions that maximize the sum of the action value $Q$, an upper confidence bound $U$ and a term $L$ that encourages programs to call programs near the same level. **2.** When the selected program is not atomic and the node has never been visited before, a new sub-tree is constructed. In the sub-tree, the LSTM internal state is initialized to zero. When the sub-tree search terminates, the LSTM internal state is reset to its previous calling state. **3.** The leaf node is expanded and the associated observation $e$ and program index $i$ are evaluated by the AlphaNPI network to compute action probabilities $P = \pi$ and values $V$. **4.** The quantities $Q$ and $U$ are computed using the network predictions. **5.** Once the search is complete, the tree policy vector $\pi^{mcts}$ is returned. The next program in the execution trace is chosen according to $\pi^{mcts}$, until the program STOP is chosen or a computational budget is exceeded.

experience tuples $(e, i, h, \pi^{mcts}, r)$ in a replay buffer. The data in this replay buffer is used to train the AlphaNPI network, as illustrated in Figure 2 .

The search approach is depicted in Figure 3 for a Tower of Hanoi example, see also the corresponding Figure 2 of Silver et al. [2017]. A detailed description of the search process, including pseudo-code, appears in Appendix A. Subsequently, we present an overview of this component of AlphaNPI.

For a specific program indexed by $i$, a node in the search tree corresponds to an observation $e$ and an edge corresponds to an action $a$. As in AlphaZero, the neural network outputs the action probabilities and node values. These values are used, in conjunction with visit counts, to compute upper confidence bounds $U$ and action-value functions $Q$ during search. Unlike AlphaZero, we add terms $L$ in the node selection stage to encourage programs not to call programs at a much lower level. In addition, we use a different estimate of the action-value function that better matches the environments considered in this paper. Actions are selected by maximizing $Q + V + L$.

Also unlike AlphaZero, if the selected action is not atomic but an already learned program, we recursively build a new Monte Carlo tree for that program. To select a trajectory in the tree, that is the program's imagined execution trace, we play $n_{simu}$ simulations and record the number of visits to each node. This enables us to compute a tree policy vector $\pi^{mcts}$ for each node, as detailed in Appendix A.5, which favours actions that have been most selected during the simulations.

The major feature of AlphaNPI is its ability to construct recursively a new tree during the search to execute an already learned program. This approach enables to use learned skills as if they were atomic actions. When a tree is initialized to execute a new program, the LSTM internal state is initialized to zero and the environment reward signal changes to reflect the specification of the new program. The root node of the new tree corresponds to the current state of the environment. When the search process terminates, we check that the final environment state satisfies the program's post-conditions. If unsatisfied, we discard the full execution trace and start again. When returning control to an

upper-level program, we assign to the LSTM the previous internal state for that level and continue the search process.

We found that discarding execution traces for programs executed incorrectly is necessary to achieve stable training. Indeed, the algorithm might choose the correct sequence of actions but still fail because one of the chosen sub-programs did not execute correctly. At the level we are trying to learn, possibly no mistake has been made, so it is wise to discard this data for training stability.

Finally, we use AlphaNPI MCTS in two different modes. In exploration mode, we use a high budget of simulations, final actions are taken by sampling according to the tree policy vectors and we add Dirichlet noise to the network priors for better exploration. This mode is used during training. In Exploitation mode, we use a low budget of simulations, final actions are taken according to the tree policy vectors argmax and we do not add noise to the priors. In this mode, AlphaNPI's behavior is deterministic. This mode is used during validation and test.

### 3.1 Training procedure

During a training iteration, the agent selects a program $i$ to learn. It plays $n_{ep}$ episodes (See Appendix E for specific values) using the tree search in exploration mode with a large budget of simulations. The generated experiences, $(e, i, h, \pi^{mcts}, r)$, where $r$ is the episode final reward, are stored in a replay buffer. The agent is trained with the Adam optimizer on this data, so as to minimize the loss function:

$$\ell = \sum_{\text{batch}} \underbrace{- \left(\pi^{mcts}\right)^T \log \pi}_{\ell_{\text{policy}}} + \underbrace{(V - r)^2}_{\ell_{\text{value}}}. \tag{2}$$

Note that the elements of a mini-batch may correspond to different tasks and are not necessarily adjacent in time. Given that the buffer memory is short, we make the assumption that the LSTM internal states have not changed too much. Thus, we do not use backpropagation through time to train the LSTM. Standard backpropagation is used instead, which facilitates parallelization.

After each Adam update, we perform validation on all tasks for $n_{val}$ episodes. The agent average performance is recorded and used for curriculum learning, as discussed in the following subsection.

### 3.2 Curriculum learning

As with previous NPI models, curriculum learning plays an essential role. As programs are organized into levels, we begin by training the agent on programs of level 1 and then increase the level when the agent's performance is higher than a specific threshold. Our curriculum strategy is similar to the one by Andreas et al. [2017].

At each training iteration, the agent must choose the next program to learn. We initially assign equal probability to all level 1 programs and zero probability to all other programs. At each iteration, we update the probabilities according to the agent's validation performance. We increase the probability of programs on which the agent performed poorly and decrease the probabilities of those on which the agent performed well. We compute scores $c_i = 1 - R_i$, for each program indexed by $i$, where $R_i$ is a moving average of the reward accrued by this program during validation. The program selection probability is then defined as a softmax over these scores. When $\min_i R_i$ becomes greater than some threshold $\Delta_{\text{curr}}$, we increase the maximum program level, thus allowing the agent to learn level 2 programs, and so on until it has learned every program.

## 4 Experiments

In the following experiments, we aim to assess the ability of our RL agent, AlphaNPI, to perform the sorting tasks studied by Reed and de Freitas [2016] and Cai et al. [2017]. We also consider a simple recursive Tower of Hanoi Puzzle. An important question we would like to answer is: Can AlphaNPI, which is trained by RL only, perform as well as the iterarive and recursive variants of NPI, which are trained with a strong supervisory signal consisting of full execution traces? Also, how essential is MCTS planning when deploying the neural network policies?

| Length | Iterative BUBBLESORT | | Recursive BUBBLESORT | |
|---|---|---|---|---|
| | Net with planning | Net only | Net with planning | Net only |
| 10 | 100% | 85% | 100% | 70% |
| 20 | 100% | 85% | 100% | 60% |
| 60 | 95% | 40% | 100% | 35% |
| 100 | 40% | 10% | 100% | 10% |

Table 1: Performance of AlphaNPI, trained on BUBBLESORT instances of length up to 7, on much longer input lists. For each BUBBLESORT variant, iterative and recursive, we deployed the trained AlphaNPI networks with and without MCTS planning. The results clearly highlight the importance of planning at deployment time.

| Length | Sorting without a hierarchy | |
|---|---|---|
| | Net with planning | Net only |
| 3 | 94% | 78% |
| 4 | 42% | 22% |
| 5 | 10% | 5% |
| 6 | 1% | 1% |

Table 2: Test performance on iterative sorting with no use of hierarchy. The AlphaNPI network is trained to sort using only atomic actions on lists of length up to 4, and tested on lists of length up to 6. The training time without hierarchy scales quadratically with list length, but only linearly with list length when a hierarchy is defined.

## 4.1 Sorting example

We consider an environment consisting of a list of $n$ integers and two pointers referencing its elements. The agent can move both pointers and swap elements at the pointer positions. The goal is to learn a hierarchy of programs and to compose them to realize the BUBBLESORT algorithm. The library of programs is summarized in Table 4 of the Appendix.

We trained AlphaNPI to learn the sorting library of programs on lists of length 2 to 7. Each iteration involves 20 episodes, so the agent can see up to 20 different training lists. As soon as the agent succeeds, training is stopped, so the agent typically sees less than 20 examples per iteration.

We validated on lists of length 7 and stopped when the minimum averaged validation reward, among all programs, reached $\Delta_{curr}$. After training, we measured the generalization of AlphaNPI, in exploitation mode, on test lists of length 10 to 100, as shown in Table 1. For each length, we test on 40 randomly generated lists.

We observe that AlphaNPI can learn the iterative BUBBLESORT algorithm on lists of length up to 7 and generalize well to much longer lists. The original NPI, applied to iterative BUBBLESORT, had to be trained with strong supervision on lists of length 20 to achieve the same generalization. As reported by Cai et al. [2017], when training on arrays of length 2, the iterative NPI with strong supervision fails to generalize but the recursive NPI generalizes perfectly. However, when training the recursive NPI with policy gradients RL and curricula, Xiao et al. [2018] reports poor results.

To assess the contribution of adding a hierarchy to the model, we trained AlphaNPI with atomic actions only to learn iterative BUBBLESORT. As reported on Table 2, this ablation performs poorly in comparison to the hierarchical solutions.

We also defined a sorting environment in which the programs RESET, BUBBLE and BUBBLESORT are recursive. This setting corresponds to the "full recursive" case of Cai et al. [2017]. Being able to learn recursive programs requires adapting environment. For instance, when a new task (recursive program) is started, the sorting environment becomes a sub-list of the original list. When the task terminates, the environment is reset to the previous list.

We trained the full recursive BUBBLESORT on lists of length 2 to 4 and validated on lists of length 7. After training, we assessed the generalization capabilities of the recursive AlphaNPI in Table 1. The results indicate that the recursive version outperforms the iterative one, confirming the results reported by Cai et al. [2017]. We also observe that AlphaNPI with planning is able to match the generalization performance of the recursive NPI with strong supervision, but that removing planning from deployment (i.e. using a network policy only) reduces performance.

| Number of disks | MCTS | Network only |
|:---:|:---:|:---:|
| 2 | 100% | 100% |
| 5 | 100% | 100% |
| 10 | 100% | 100% |
| 12 | 100% | 100% |

Table 3: Test performance of one AlphaNPI trained agent on the recursive Tower of Hanoi puzzle.

## 4.2 Tower of Hanoi puzzle

We trained AlphaNPI to solve the Tower of Hanoi puzzle recursively. Specifically, we consider an environment with 3 pillars and $n$ disks of increasing disk size. Each pillar is given one of three roles: source, auxiliary or target. Initially, the $n$ disks are placed on the source pillar. The goal is to move all disks to the target pillar, never placing a disk on a smaller one. It can be proven that the minimum number of moves is $2^n - 1$, which results in a highly combinatorial problem. Moreover, the iterative solution depends on the parity of the number of disks, which makes it very hard to learn a general iterative solution with a neural network.

To solve this problem recursively, one must be able to call the TOWEROFHANOI program to move $n - 1$ disks from the source pillar to the auxiliary pillar, then move the larger disk from the source pillar to target pillar and finally call again the TOWEROFHANOI program to move the $n - 1$ pillars from the auxiliary pillar to the target.

We trained our algorithm to learn the recursive solution on problem instances with 2 disks, stopping when the minimum of the validation average rewards reached $\Delta_{\text{curr}}$. Test results are shown in Table 3. AlphaNPI generalizes to instances with a greater number of disks.

In Appendix C, we show that once trained, an AlphaNPI agent can generalize to Tower of Hanoi puzzles with an arbitrary number of disks.

## 5 Related work

AlphaZero [Silver et al., 2017] used Monte Carlo Tree Search for planning and to derive a policy improvement operator to train state-of-the-art neural network agents for playing Go, Chess and Shogi using deep reinforcement learning. In [Laterre et al., 2018], AlphaZero is adapted to the setting of one-player games applied to the combinatorial problem of bin packing. This work casts program induction as a one player game and further adapts AlphaZero to incorporate compositional structure into the learned programs.

Many existing approaches to neural program induction do not explicitly learn programs in symbolic form, but rather implicitly in the network weights and then directly predict correct outputs given query inputs. For example, the Neural GPU [Kaiser and Sutskever, 2015] can learn addition and multiplication of binary numbers from examples. Neural module networks [Andreas et al., 2016] add more structure by learning to stitch together differentiable neural network modules to solve question answering tasks. Neural program meta induction [Devlin et al., 2017a] shows how to learn implicit neural programs in a few-shot learning setting.

Another class of neural program induction methods takes the opposite approach of explicitly synthesizing programs in symbolic form. DeepCoder [Balog et al., 2016] and RobustFill [Devlin et al., 2017b] learn in a supervised manner to generate programs for list and string manipulation using domain specific languages. In [Evans and Grefenstette, 2018], explanatory rules are learned from noisy data. Ellis et al. [2018] shows how to generate graphics programs to reproduce hand drawn images. In [Sun et al., 2018], programs are generated from visual demonstrations. Chen et al. [2017] shows how to learn parsing programs from examples and their parse trees. Verma et al. [2018] shows how to distill programmatically-interpretable agents from conventional Deep RL agents.

Some approaches lie in between fully explicit and implicit, for example by making execution differentiable in order to learn parts of programs or to optimize programs [Bošnjak et al., 2017, Bunel et al., 2016, Gaunt et al., 2016]. In [Nye et al., 2019], an LSTM generator conditioned on specifications is used to produce schematic outlines of programs, which are then fed to a simple

logical program synthesizer. Similarly, Shin et al. [2018] use LSTMs to map input-output pairs to traces and subsequently map these traces to code.

Neural Programmer-Interpreters [Reed and de Freitas, 2016], which we extend in this work, learn to execute a hierarchy of programs from demonstration. Cai et al. [2017] showed that by learning recursive instead of iterative forms of algorithms like bubble sort, NPI can achieve perfect generalization from far fewer demonstrations. Here, perfect generalization means generalization with provable theoretical guarantees. Neural Task Programming [Xu et al., 2018] adapted NPI to the setting of robotics in order to learn manipulation behaviors from visual demonstrations and annotations of the program hierarchy.

Several recent works have reduced the training data requirements of NPI, especially the "strong supervision" of demonstrations at each level of the program hierarchy. For example, Li et al. [2017] and Fox et al. [2018] show how to train variations of NPI using mostly low-level demonstration trajectories and a relatively smaller proportion of hierarchical annotations compared to NPI. However, demonstrations are still required. Xiao et al. [2018] incorporates combinator abstraction techniques from functional programming into NPI to improve training, but emphasize the difficulty of learning simple NPI models with RL algorithms.

Hierarchical reinforcement learning combined with deep neural networks has received increased attention in the past several years [Osa et al., 2019, Nachum et al., 2018b, Kulkarni et al., 2016, Nachum et al., 2018a, Levy et al., 2018, Vezhnevets et al., 2017], mainly applied to efficient training of agents for Atari, navigation and continuous control. This work shares a similar motivation of using hierarchy to improve generalization and sample efficiency, but we focus on algorithmic problem domains and learning potentially recursive neural programs without any demonstrations.

While AlphaZero does not use hierarchies or recursion, hierarchical MCTS algorithms have been previously proposed for simple hierarchical RL domains [Vien and Toussaint, 2015, Bai et al., 2016]. The current work capitalizes on advances brought in by deep reinforcement learning as well as design choices particular to this paper to significantly extend this research frontier.

Finally, as demonstrated in the original NPI paper, the modular approach with context-dependent input embeddings and a task independent interpreter is ideal for meta-learning and transfer. Recent manifestations of this idea of using an embedding to re-program a core neural network to facilitate meta-learning include Zintgraf et al. [2019] and Chen et al. [2019]. To the best of our knowledge the idea of programmable neural networks goes back several decades to the original Parallel Distributed Programming (PDP) papers of Jay McClelland and colleagues. We leave transfer and meta-learning as a future explorations for AlphaNPI.

# 6   Conclusion

This paper proposed and demonstrated the first effective RL agent for training NPI models: AlphaNPI. AlphaNPI extends NPI to the RL domain and enhances AlphaZero with the inductive biases of modularity, hierarchy and recursion. AlphaNPI was shown to match the performance of strongly supervised versions of NPI in the sorting experiment, and to generalize remarkably well in the Tower of Hanoi environment. The experiments also shed light on the issue of deploying neural network RL policies. Specifically, we found out that agents that harness MCTS planning at test time are much more effective than plain neural network policies.

While our test domains are complex along some axes, e.g. recursive and combinatorial, they are simple along others, e.g. the environment model is available. The natural next step is to consider environments, such as robot manipulation, where it is also important to learn perception modules and libraries of skills in a modular way to achieve transfer to new tasks with few data. It will be fascinating to harness imperfect environment models in these environments and assess the performance of MCTS planning when launching AlphaNPI policies.

# 7   Acknowledgements

Work by Nicolas Perrin was partially supported by the French National Research Agency (ANR), Project ANR-18-CE33-0005 HUSKI.

## Footnotes

[1]The code is available at `https://github.com/instadeepai/AlphaNPI`

[2]A detailed description of AlphaNPI is provided in Appendix A.

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
