[Supplementary Material]

# Appendices

## A    Detailed description of AlphaNPI

When learning the $i$-th program, the environment is reset to a state that satisfies the program pre-conditions. At the same time, we adopt a reward that returns 1 when the program post-condition is satisfied and 0 otherwise. A tree is built for this specific task, to maximize the expected reward.

For a specific program, a node corresponds to an environment state[3] $e$ and an edge corresponds to an action $a$. The root node corresponds to the initial environment state. Every node contains a prior, a visit count and a Q-value estimate. The prior $P(e, a)$ corresponds to the probability of this node being chosen by its parent node. The visit count $N(e, a)$ records how many times this node has been visited through simulations and the Q-value $Q(e, a)$ estimates the expected reward the agent will accrue if it chooses action $a$ from this node.

A simulation involves three operations: select, expand and evaluate, and value backup. When a selected action corresponds to a non-zero level program, we recursively build a new tree to execute it, see Algorithm 1. Finally, when a given budget of simulations has been spent, a tree policy vector is computed and the next action is chosen according to this vector. We delve into the details of these steps in the following subsections. These steps are illustrated in Figure 3.

### A.1    Select

From a non-terminal node, the next action is chosen to maximise the P-UCT criterion:

$$a = \operatorname*{argmax}_{a' \in \mathcal{A}} \left( Q(e, a') + U(e, a') + L(i, a') \right), \tag{3}$$

$$U(e, a) = c_{\text{puct}} P(e, a) \frac{\sqrt{\sum_b N(e, b)}}{1 + N(e, a)}. \tag{4}$$

The coefficient $c_{\text{puct}}$ is user-defined and trades-off exploration and exploitation. The level balancing term $L$ is defined as:

$$\begin{cases} L(i, a) = c_{\text{level}} \exp(-1), & \text{if } a \text{ is STOP} \\ L(i, a) = c_{\text{level}} \exp(-1), & \text{if level}(i) = \text{level}(a) \\ L(i, a) = c_{\text{level}} \exp(-(\text{level}(i) - \text{level}(a))), & \text{otherwise} \end{cases} \tag{5}$$

where $c_{\text{level}}$ is a user-defined constant and level is an operator that returns a program level. This term encourages programs to call programs near them in the hierarchy.

We perform additional exploration as in the original AlphaZero work of Silver et al. [2017] by adding Dirichlet noise to the priors:

$$P(e, a) \longleftarrow (1 - \epsilon_d) P(e, a) + \epsilon_d \eta_d, \quad \text{where } \eta_d \sim \text{Dir}(\alpha_d) \tag{6}$$

where $\eta_d$ follows a Dirichlet distribution with hyper-parameters $\alpha_d$.

### A.2    Tree recursion

If the chosen action is atomic, we apply it directly in the environment and the new environment observation is recorded inside the node. Otherwise, a new tree is built to execute the newly invoked program. In this case, the environment reward changes to correspond to this new task (program), and the LSTM internal state is re-initialized to zero. The new tree is built in exploitation mode. When the search terminates, we check if the program post-conditions are satisfied. If unsatisfied, we stop the entire search and discard the corresponding trace. If satisfied, the task at hand becomes the previous one (return to calling program). In this case, the LSTM is assigned its previous internal state and the new environment state is recorded inside the child node. From this point of view, the program has been executed as if it was an atomic action.

**Algorithm 1:** Perform one MCTS simulation
---
**Input**: Node $n = (e, i)$ and LSTM internal state $h$
**while** *True* **do**
    **if** *n has not been expanded* **then**
        Compute possible children nodes to respect programs levels and pre-conditions;
        Evaluate the node with NPI network to compute priors and V-value;
        If the mode is exploration add Dirichlet noise to the priors;
        Get and store new LSTM internal state $h$;
        Store the priors in the node;
        Stop the simulation and return V-value;
    **else**
        Select an action $a$ to according to Equation 3;
        **if** *simulation length $\geq$ maximum tree depth* **then**
            Stop the simulation ;
            Return a reward of -1
        **else**
            **if** $a == STOP$ **then**
                Stop the simulation ;
                Return the obtained reward
            **else**
                **if** *a is a level 0 program* **then**
                    Apply $a$ in the environment[4];
                **else**
                    Build a new tree[4] in exploitation mode to execute $a$ ;
                **end**
                Record new environment observation $e'$;
                Consider new node $n = (e', i)$;
            **end**
        **end**
    **end**
**end**

## A.3 Expand and evaluate

When a node is expanded, we construct a new child node for every possible action available to the parent node. The possible actions correspond to the programs whose pre-conditions are satisfied and whose level is lower (or equal if the setup is recursive) than the current program's level. The priors over the edges and node V-value are computed using the AlphaNPI network. The child nodes' Q-values are initialized to 0 and their visit counts to 0.

## A.4 Value back-up

When an action is chosen, if the new node is not terminal, we compute its value with the value head of the AlphaNPI network. Otherwise, we use the episode's final reward. When the obtained reward equals 0, we replace it by a value of -1 as the Q-values are initialized to 0. To encourage short traces, we penalize positive rewards by multiplying them by $\gamma^n$, where $n$ is the trace length and $\gamma \in [0, 1]$. This value is then backpropagated on the path between the current node and the root node. The visit counts of the nodes in this path are also incremented. For each node, we maintain a list of all the values that have been backpropagated. In classical two-player MCTS, the Q-value is computed as the sum of the values, generated by the neural network for the child node, divided by the visit count.

**Algorithm 2:** AlphaNPI tree search

---

**Input**: program index $i$ and a mode (exploration/exploitation)
Initialize LSTM internal state $h$ to 0
Initialize execution trace to empty list
Get initial environment observation $e_0$
Build root node $n = (e_0, i)$
**while** $True$ **do**
    **for** k = 1, ..., $n_{simu}$ **do**
        Reset the environment in the state corresponding to root node $n$
        Perform one simulation from root node $n$ using Algorithm 1
        Back-up values in the tree
        Update nodes visit counts
    **end**
    Compute tree policy $\pi^{mcts}$ with visit counts
    Choose next action $a \sim \pi^{mcts}$ (exploration) or $a = \text{argmax}\, \pi^{mcts}$ (exploitation)
    Add action to the execution trace
    **if** *trace length $\geq$ maximum tree depth* **then**
        Stop the search;
        Return the execution trace and a -1 reward;
    **else**
        **if** $a == STOP$ **then**
            Stop the search;
            Get final reward;
            Return the execution trace and the final reward;
        **else**
            From root node select the edge corresponding to $a$;
            The outgoing node becomes the new root node $n$;
        **end**
    **end**
**end**

---

Since our approach is single-player, we use a slightly different expression:

$$
\begin{aligned}
Q(e,a) &= \sum_{e'|e,a\to e'} p_{e'} V(e'), \\
p_{e'} &= \frac{\exp\left(\tau_1 V(e')\right)}{\sum_{e'|e,a\to e'} \exp\left(\tau_1 V(e')\right)},
\end{aligned}
\tag{7}
$$

where $\tau_1$ is a temperature coefficient and $e'|e, a \to e'$ indicates that a simulation eventually reached $e'$ after taking action $a$ from node $e$. In two-player MCTS, the expected reward does not depend only on the chosen action but also on the other player's response. Due to the stochasticity (typically adversarial) of this response, it is judicious to choose actions with good Q-values on average. In our single-player approach, the environment is deterministic, therefore we focus on a highly rewarding course of actions.

## A.5   Final execution trace construction

To compute the final execution trace, we begin at the tree root-node and launch $n_{simu}$ simulations. We choose the next action in the execution trace according to the nodes' visit counts. We compute tree policy vectors

$$
\pi^{mcts}(a) = \frac{N(e,a)^{\tau_2}}{\sum_b N(e,b)^{\tau_2}},
\tag{8}
$$

where $\tau_2$ is a temperature coefficient. If the tree is in exploration mode, the next action is sampled according to this probability vector. If the tree is in exploitation mode, it is taken as the tree policy argmax. When an action is chosen, the new node becomes the root-node, and $n_{simu}$ simulations are played from this node and so-on until the end of the episode, see Algorithm 2. The final trajectory is then stored inside a replay buffer.

### A.6 Prioritized replay buffer

The experience generated by MCTS is stored inside a prioritized replay buffer. This experience takes the form of tuples $(e, i, h, \pi^{mcts}, r)$ where $e$ is an environment observation, $i$ the program index, $h$ the LSTM internal state, $\pi^{mcts}$ the tree policy for the corresponding node and $r$ the reward obtained at the end of the trajectory. The buffer has a maximum memory size $n_{buf}$. When the memory is full, we replace a past experience with a new one. To construct a training batch, we sample buffer tuples with probability $p_{buf}$, which measures the chance that the tuple results in positive reward. We also make sure that the buffer does not contain experiences related to tasks for which a positive reward has not been found yet.

### A.7 Network architecture

To adapt the original NPI algorithm to our setting, me make the following modifications:

1. Programs do not accept arguments anymore. The program ACT and its finite set of possible arguments are replaced by atomic actions, one for each argument.
2. The termination scalar returned by the network is replaced by the action STOP.
3. For simplicity, the program keys matrix has been replaced by a dense layer with identity activation function.
4. We added policy and value modules to the architecture to obtain an actor-critic architecture necessary for RL.

In our architecture, the LSTM core has one layer of $H$ neurons. Both the actor and the critic modules are multi-layer perceptrons with one hidden layer of $H/2$ neurons and ReLu activation functions. The encoder is environment dependent, however in the three environments we consider, it is a simple multi-layer perceptron with one hidden layer.

### A.8 Curriculum learning

In the curriculum scheduler, we maintain a maximum program level $l_{max}$, which is initialized to 1. At the beginning of each training iteration, the curriculum scheduler is called to determine the next program to be learned. This program must have a level lower than $l_{max}$. Each time a validation episode for the $i$-th program is conducted, we record the final reward $r$ and update the $i$-th program average reward $R_i$ as follows:

$$R_i \longleftarrow \beta R_i + (1 - \beta)r \tag{9}$$

where $\beta$ is a user-defined coefficient.

When the minimum average reward $R_i$ over the programs of level lower than $l_{max}$ reaches $\Delta_{\text{curr}}$ we increment $l_{max}$. To determine which program should be learned next, the curriculum scheduler computes probabilities over the programs of level lower than its internal maximum level. The $i$-th program probability $p_i$ is defined as follows

$$p_i = \frac{\exp\left(\tau_3 c_i\right)}{\sum\limits_{k} \exp\left(\tau_3 c_k\right)} \tag{10}$$
$$c_i = {}^1\!/R_i$$

where $\tau_3$ is a temperature coefficient.

## B Environments

### B.1 Sorting environment

We consider a list of $n$ digits and two pointers that refer to elements in the list. The atomic actions are moving the pointers and swapping elements at the pointer locations. The level 1 programs may move both pointers at the same time and conditionally swap elements. Level 2 programs are RESET and BUBBLE. RESET moves both pointers to the extreme left of the list and BUBBLE conditionally compares two by two the elements from left to right. BUBBLESORT is level 3 and sorts the list.

| program | description | level |
|---|---|---|
| BUBBLESORT | sort the list | 3 |
| RESET | move both pointers to the extreme left of the list | 2 |
| Bubble | make one pass through the list | 2 |
| RSHIFT | move both pointers once to the right | 1 |
| LSHIFT | move both pointers once to the left | 1 |
| COMPSWAP | if both pointers are at the same position, move pointer 2 to the left, then swap elements at pointers positions if left element > right element | 1 |
| PTR_2_L | move pointer 2 to the left | 0 |
| PTR_1_L | move pointer 1 to the left | 0 |
| PTR_1_R | move pointer 1 to the right | 0 |
| PTR_2_R | move pointer 2 to the right | 0 |
| SWAP | swap elements at the pointers positions | 0 |
| STOP | terminates current program | 0 |

Table 4: Program library for the list sorting environment.

| program | pre-condition |
|---|---|
| BUBBLESORT | both pointers are at the extreme left of the list |
| RESET | both pointers are not at the extreme left of the list |
| BUBBLE | both pointers are at the extreme left of the list |
| RSHIFT | both pointers are not at the extreme right of the list |
| LSHIFT | both pointers are not at the extreme left of the list |
| COMPSWAP | pointer 1 is directly at the left of pointer 2, or they are at the same position |
| PTR_2_L | pointer 2 is not at the extreme left of the list |
| PTR_1_L | pointer 1 is not at the extreme left of the list |
| PTR_1_R | pointer 1 is not at the extreme right of the list |
| PTR_2_R | pointer 2 is not at the extreme right of the list |
| SWAP | the pointers are not at the same position |
| STOP | no condition |

Table 5: Program pre-conditions for the list sorting environment.

In this environment, compositionality is mandatory for the tree search to find a way to sort the list when $n$ is greater than 3. Indeed, BUBBLE requires $3n$ atomic actions and RESET $2n$ atomic actions. When both programs are known, BUBBLESORT simply alternates BUBBLE and RESET $n$ times. Therefore, if BUBBLESORT had to use atomic actions only, it would require $n \times 3n + n \times 2n = 5n^2$ actions, while it might require only a correct sequence of $2n$ actions if BUBBLE and RESET programs have already been learned.

The environment observations have the form $e = (v_1, v_2, b_{1i}, b_{1e}, b_{2i}, b_{2e}, b_{12}, b_s)$ where $v_1$ and $v_2$ are the one-hot-encoded vectors that represent the digits referenced by the pointers 1 and 2. $b_1i$, $b_2i$ and $b_1e$, $b_2e$ respectively equal 1 if the pointer 1/2 is at the beginning/end of the list and 0 otherwise. $b_{12}$ equals 1 if both pointers are at the same position and 0 otherwise. $b_s$ equals 1 if the list is sorted and 0 otherwise. The dimension of the observation space is 26. The encoder is composed of one hidden layer with 100 neurons and a ReLu activation function.

The program library specification appears in Table 4, with the pre-conditions defined in Table 5.

## B.2 Recursive Sorting environment

We consider the same environment and the same programs library than for the non-recursive case. The only difference is the environment ability to decrease the size of the list when a task corresponding to recursive program starts and to increase back its size when the task ends.

Learning recursive programs in this environment has the strong advantage to remove the dependency to execution traces length. Indeed, in the non-recursive case, the size of the execution traces of Reset,

Bubble and Bubblesort depends linearly of the list length. Their execution traces size are constant in the recursive case which facilitates the tree search.

## B.3 Tower of Hanoi environment

We consider three pillars and $n$ disks. When a game is started, each pillar is attributed an initial role that can be source, auxiliary or target. The $n$ disks are initially placed on the source pillar in decreasing order, the largest one being at the bottom. The goal is to move all disks from the source pillar to the target pillar, without ever placing a disk on a smaller one. For each pillar, we consider its initial role and its current role. At the beginning, both are equivalent. Acting consists of switching the current roles of two pillars and moving a disk from the current source pillar to the current target pillar.

The game starts when the program TOWEROFHANOI is called. If during a game the TOWEROFHANOI program is called again, i.e. is called recursively, the largest disk is removed and the game restarts. The roles of the initial pillars become the current roles in the previous game. The reward signal changes accordingly. When TOWEROFHANOI terminates, the largest disk is placed back at its previous location and the pillars get the initial roles they had in the previous game.

Figure 4: Tower of Hanoi environment illustration.

The combinatorial nature of the Tower of Hanoi puzzle, and in particular its sparse reward signal, makes this game a challenge for conventional reinforcement learning algorithms. In [Edwards et al., 2018], the authors introduced a backward induction, to enable the agent to reason backwards in time. Through an iterative process, it both explores forwards from the start position and backwards from the target/goal. They have shown that by endowing the agent with knowledge of the reward function, and in particular of the goal, it can outperform the standard DDQN algorithm. However, their experiments were limited to three-disks which they solved perfectly without mentioning any generalisation performance beyond this number.

The environment observations have the form $e = (m_1, m_2, m_3, b_n, b_s)$ where $m_1$, $m_2$ and $m_3$ are equal to 1 if respectively the move from the source/auxiliary/source pillar to the auxiliary/target/target pillar is possible, and 0 otherwise. $b_n$ equals 1 if $n = 1$ and 0 otherwise. $b_s$ equals 1 if the puzzle is solved, i.e. all the disks are on the target pillar and the target pillar is in its initial location. Therefore, the observations dimension is 5. The encoder is composed of one hidden layer with 100 neurons and a ReLu activation function.

The program library specification appears in Table 6, while the pre-conditions are defined in Table 7.

| program | description | level |
|---|---|---|
| TOWEROFHANOI | move $n$ disks from source pillar to target pillar | 1 |
| SWAP_S_A | source pillar becomes auxiliary and vice-versa | 0 |
| SWAP_A_T | auxiliary pillar becomes target and vice-versa | 0 |
| MOVEDISK | move disk from source to target | 0 |
| STOP | terminates current program | 0 |

Table 6: Program library for Tower of Hanoi.

| program | pre-condition |
|---|---|
| TOWEROFHANOI | all $n$ disks are on the source pillar |
| SWAP_S_A | the number of disks is greater than one |
| SWAP_A_T | the number of disks is greater than one |
| MOVEDISK | the move from source to target is possible |
| STOP | no pre-condition |

Table 7: Program pre-conditions for Tower of Hanoi.

The TOWEROFHANOI post-condition is satisfied when $n$ disks have been moved from the initial source pillar to the initial target pillar and all pillars' current roles correspond to their initial roles. When TOWEROFHANOI is called recursively, its pre-condition is tested in the new environment with the largest disk removed.

## C   Tower of Hanoi recursion proof

In this section, we prove that once trained AlphaNPI can generalize to Hanoi puzzles with an arbitrary number of disks.

We remind the reader that the environment observations have the form $e = (m_1, m_2, m_3, b_n, b_s)$ where $m_1$, $m_2$ and $m_3$ are equal to 1 if respectively the move from the source/auxiliary/source pillar to the auxiliary/target/target pillar is possible, and 0 otherwise. $b_n$ is equal to 1 if $n = 1$ and 0 otherwise. $b_s$ is equal to 1 if the puzzle is solved, i.e. all the disks are on the target pillar and all pillars are at their initial locations. Otherwise, $b_s = 0$.

The environment is initialized with all disks on the source pillar. Therefore, there are only two possible initial observations: $e_0^1 = (1, 0, 1, 1, 0)$ if there is only 1 disk, and $e_0^n = (1, 0, 1, 0, 0)$ if $n \geq 2$.

In exploitation mode, AlphaNPI has a deterministic behavior, so two identical sequences of observations necessarily correspond to the exact same sequence of actions.

We assume that the trained agent solves the case $n = 1$, and that, for $n = 2$, it solves the Hanoi puzzle with the following sequence of observations and actions:

- $e_0^2 = (1, 0, 1, 0, 0) \rightarrow$ SWAP_A_T
- $e_1^2 = (1, 0, 1, 0, 0) \rightarrow$ TOWEROFHANOI
- $e_2^2 = (1, 0, 0, 0, 0) \rightarrow$ SWAP_A_T
- $e_3^2 = (0, 1, 1, 0, 0) \rightarrow$ MOVEDISK
- $e_4^2 = (0, 1, 0, 0, 0) \rightarrow$ SWAP_S_A
- $e_5^2 = (1, 0, 1, 0, 0) \rightarrow$ TOWEROFHANOI
- $e_6^2 = (0, 0, 0, 0, 1) \rightarrow$ SWAP_S_A
- $e_7^2 = (0, 0, 0, 0, 1) \rightarrow$ STOP

For any $n \geq 3$, the initial observation is the same: $e_0^n = e_0^2 = (1, 0, 1, 0, 0)$, leading to the same action SWAP_A_T. The next observation is again $(1, 0, 1, 0, 0)$, so the second action is also TOWEROFHANOI. Assuming that the recursive call to TOWEROFHANOI is successful (i.e. the case $n - 1$ is solved), it can be verified that the exact same sequence of 8 observations and actions is generated. Besides, if the recursive call to TOWEROFHANOI is successful, this sequence of 8 actions actually solves the puzzle. By induction, we conclude that for any $n \geq 2$, the agent generates the same sequence of actions, which solves the puzzle.

This proof shows that by simply observing the behavior of the trained agent on the cases with 1 and 2 disks, we can possibly acquire the certainty that the agent generalizes correctly to any number of disks. By encouraging agents to try recursive calls during their training (see Section E.1), AlphaNPI agents often end up solving the case $n = 2$ with the above sequence of actions. So, even though this generalization proof does not apply to every trained agent, it is often a convenient way to verify the correctness of the agent's behavior for any number of disks.

# D  More experimental results and comparisons

Figure 5: Validation reward evolution during training on the sorting environment for the different programs in the library. The TOTAL curve corresponds to the sum of all programs rewards divided by the number of programs.

Figure 5 shows how the progress on non-elementary programs starts once a good performance has been reached with all the programs of lower level in the hierarchy. It can be observed that, for the BUBBLESORT program, the performance converges towards almost 100% of success after approximately 250 iterations, which corresponds to the training on $250 \times 20 = 5000$ traces (20 episodes are played at each iteration). In the original NPI work, the network was trained on 1216 BUBBLESORT execution traces, which represents much more data as each trace contains also sub-traces corresponding to all the executed sub-programs, while in our setting traces are limited to the actions of a single program. Yet, our results exhibit better generalization properties.

With our method, the sizes of instances are limited but random during training. There is no ordering of instances, as the curriculum uses only the hierarchy of programs. This hierarchy is not explicit in Cai et al. [2017] and in Reed and de Freitas [2016], but it could be easily inferred from the observed execution traces. In Xiao et al. [2018], authors report that learning programs via reinforcement learning was not successful, even with a curriculum. To facilitate the reinforcement learning, they used an adaptive sampling technique proposed in Reed and de Freitas [2016] to fetch example traces with a frequency proportional to the current prediction error. In our work, traces are never given in advance, they are generated via interactions between the system and the environment. Since execution traces somehow require the problem to be already solved, replacing them by a hierarchy of programs is, in our opinion, a significantly weaker supervision requirement.

# E  Implementation details

The code has been developed in Python3.6. We used Pytorch as the Deep Learning library. The code is not based on existing implementations. Our AlphaNPI architecture, the search algorithm and the environments have been developed from scratch. The code is open-source.

## E.1  Recursive programs

When specifying a program library, the user can define recursive programs. In this setting, a program can call lower level programs and itself. When the program calls itself, a new tree is recursively built to execute it and everything happens as for any other program call. As we train the recursive programs on small problem instances, for example on small lists, MCTS is likely to find non-recursive solutions.

Hence, when a program is defined as recursive by the user, we encourage the algorithm to find a recursive execution trace, i.e. an execution in which the program calls itself, by subtracting from the reward of non-recursive programs a constant $r_{pen-recur}$. Note that this helps the algorithm find recursive solutions, but does not completely prevent it from finding non-recursive ones.

## E.2 Computation resources

We trained AlphaNPI on the three environments with a 12 CPUs laptop with no GPU. With this computing architecture, training on one environment (Tower of Hanoi or BUBBLESORT) takes approximately 2 hours.

# F Hyper-parameters

| Notation | description | value |
|---|---|---|
| $P$ | program embedding dimension | 256 |
| $H$ | LSTM hidden state dimension | 128 |
| $S$ | observation encoding dimension | 32 |
| $\Delta_{curr}$ | threshold in curriculum learning | 0.97 |
| $\gamma$ | discount factor to penalize long traces reward | 0.97 |
| $n_{simu}$ | number of simulations performed in the tree in exploration mode | 200/1500[5] |
| $n_{simu-exploit}$ | number of simulations performed in the tree in exploitation mode | 5 |
| $n_{batches}$ | number of batches used for training at each iteration | 2 |
| $n_{ep}$ | number of episodes played at each iteration | 20 |
| $n_{val}$ | number of episodes played for validation | 25 |
| $c_{level}$ | coefficient to encourage choice of higher level programs | 3.0 |
| $c_{puct}$ | coefficient to balance exploration/exploitation in MCTS | 0.5 |
| *batch size* | batch size | 256 |
| $n_{buf}$ | buffer memory maximum size | 2000 |
| $p_{buf}$ | probability to draw positive reward experience in buffer | 0.5 |
| $lr$ | learning rate | 0.0001 |
| $r_{pen-recur}$ | penalty to encourage recursive execution trace | 0.9 |
| $\tau_1$ | q-values computation temperature coefficient | 1.0 |
| $\tau_2$ | tree policies temperature coefficient | 1.3 |
| $\tau_3$ | curriculum temperature coefficient | 2.0 |
| $\beta$ | curriculum scheduler moving average | 0.99 |
| $\epsilon_d$ | AlphaZero Dirichlet noise fraction | 0.25/0.5[5] |
| $\alpha_d$ | AlphaZero Dirichlet distribution parameter | 0.03/0.5[5] |

Table 8: Hyperparameters.

## Footnotes

[3]Strictly speaking, the node is for the pair $(e, i)$ but we are assuming a fixed $i$ and dropping the program index to simplify notation.

[4]When an action $a$ is called, we apply it into the environment only the first time. The resulting environment state is stored inside the outgoing node. When this action is to be called again from the same node we reset the environment in the correct environment state. We apply the same strategy when $a$ corresponds to a non-atomic program. The recursive tree is built only the first time.

[5]We use a greater number of simulations and add more Dirichlet noise to train AlphaNPI on the Tower of Hanoi than to train it on BUBBLESORT because of the higher complexity of the problem.