[Reviews · NeurIPS 2019]

Reviewer 1



## Summary The paper extends the NPI formalism studied by Reed & de Freitas and later by Cai et al., to be learnable by an AlphaZero-like reinforcement learning without the need for strong supervision with execution traces. It instead learns the hierarchy of program subroutines in a curriculum fashion, adding a pre- and post-condition to each subroutine and extending the MCTS setup of AlphaZero to handle recursive subroutine calls. The paper demonstrates that the resulting formulation learns the programs in both Sorting and TowersOfHanoi domains more effectively than prior work. ## Review The main contribution of the paper is generalization of the NPI setting to eliminate the need for strong supervision. This required innovation in both components of the final solution: MCTS and NPI. The experimental results are commendable, clearly showing much stronger generalization of the learned NPI models even without as strong of supervision as the baselines required. However, this is to some extent a false equivalence. The requirement for strong supervision has been replaced with the requirement for curriculum learning with a clearly marked hierarchy of subroutines including pre- and post-conditions. I was surprised by the remark in §3.2 that prior approaches also relied on curriculum learning. While Cai et al. trained their NPI system on much smaller instances than at test time, it was for generalization evaluation rather than curriculum learning. The instances weren't ordered and there were no hyperparameters accounting for the agent's progress (like Δ_curr in this work). So at a high level, strong supervision of one kind was traded for strong supervision of another. In addition to the main results, I found Table 2 a super interesting experiment. I would cast its results in a different light than the authors. It seems to show that even without curriculum learning and a hierarchy of subroutines, training weakly supervised NPI models with reinforcement learning using AlphaZero provides an improvement (even if the setting remains impractically hard anyway). This is an important result in and of itself, and I would devote some space investigating it in the final version of the paper. For instance, how do the results change over the course of training? What happens if we a little bit of supervision (e.g. a single level of recursive subroutines)? ## Minor remarks * Would be great to include baseline results of Cai et al. and Xiao et al. into Tables 1-2 to appreciate the numbers in context. * The (shortened version of) Algorithms 1-2 with some commentary might be much more helpful for explaining AlphaNPI than the current lines 120-150. ------------------------------------------------------------ ## Post-response update I'd like to thank the authors for their answers and clarifications. My score remains a positive "7": I believe this work is already a good contribution to NeurIPS. Its presentation clarity should be improved for camera-ready, and the authors have clearly outlined all the improvements they intend to integrate.

Reviewer 2



The authors apply an actor critic algorithm to the task of learning to solve algorithmic/program induction tasks. The model uses an LSTM and embeddings of program states and actions, with a shared policy and value network. During rollouts, MCTS is used as an improvement operator. Following prior work on curriculum learning, the authors check the validation error periodically and adjust using this the task regiment, increasing in difficulty over time. The authors validate their approach by training to learn how to sort, and solve a Tower of Hanoi problem, they demonstrate that using a recursive solution, it is possible to train a policy to solve either of these tasks without full trace supervision.

Reviewer 3



** update after author response ** The author response is lucid and to the point. My concerns are cleared. So I decide to raise the score to 8. This paper presents a method AlphaNPI to learn neural programs without groundtruth execution traces. Instead, post-condition tests are used to provide reward signals. AlphaNPI could try different atomic actions and novel subprograms, and reuse subprograms that leads to high rewards. It's verified to generalize well on bubblesort and tower of hanoi. This could be a very good work. However I feel a few points should be improved: 1. The idea of how a new subprogram is learned and added to M_prog is not clearly presented, which I think is the core contribution of this paper. 2. The experiments are a bit weak. Basically the authors compare the model with its own variants. In addition, as in typical RL papers, the authors should provide the training iterations and reward curves, so that readers will have an idea of its sample efficiency. 3. AlphaNPI uses post-conditions to make sure a learned subprogram brings direct benefits. This is interesting, but I'm also worried that this may be limiting. In more sophisticated problems, not every subprogram can be measured this way. Some subprograms could be utility functions and do not provide direct rewards. It's difficult to define meaningful post-conditions for such subprograms. I guess that's why the authors choose bubblesort and tower of hanois (because they can be decomposed into easier subtasks, and post-conditions can measure whether each subtask is accomplished successfully.) Though, this work could be an interesting initial attempt along this line. 4. How the tree policy vector pi_mcts encodes the tree? It's not explained.

[Author Response · NeurIPS 2019]

We warmly thank the reviewers for their careful reading of the paper and for their feedback which will help us to significantly improve the final version. We agree with the reviewers that we can present our work more clearly, that some additional experiments will provide valuable information and that we can give a more accurate perspective about our future work and the current limitations of AlphaNPI.

# 1 Clarity

We thank the reviewers for their insightful suggestions to improve the clarity of our paper.

In Section 3, we will focus on clarifying the key principles of the algorithm, and the way the hierarchies of programs are exploited. We will add a short description of Algorithms 1 and 2, the MCTS simulation and tree search, to improve readability without necessarily referring to the appendix. Besides, as requested by Reviewer #2 , we have already worked on a new figure that will illustrate better the main mechanisms of AlphaNPI.

We will also add a paragraph to recap with more details the ways the $\pi^{mcts}$ output is computed and used. In a nutshell, for each observation $e$, the $\pi^{mcts}$ vector, computed with AlphaZero via a guided MCTS simulation, corresponds to a distribution over actions. This is assumed to be better than the network output for this same observation, where better means closer to the one generated by an optimal policy. Therefore, for all observations in the generated traces, we minimize a distance between the distribution represented by $\pi^{mcts}$ and the current network output distribution to adjust the model weights. The exact computation of the $\pi^{mcts}$ vector can be found in Appendix A.5, which we will improve.

The comments of the reviewers also helped us to realize that the role of $M_{prog}$ could be better explained. We will clarify the training and use of program embeddings, and the slight differences between our work and the original NPI.

Finally, Reviewer #1 comments that, compared to the work of Cai et al., a previous extension of NPI, we have traded strong supervision of one kind for strong supervision of another. However, we would like to clarify that with our method, the sizes of instances are limited but random during training. There is no ordering of instances, as the curriculum uses only the hierarchy of programs. This hierarchy is not explicit in Cai et al. and in Reed & de Freitas, but it could be easily inferred from the observed execution traces. In our work, traces are not given in advance, they are generated via interactions between the system and the environment. Since execution traces somehow require the problem to be already solved, replacing them by a hierarchy of programs is, in our opinion, a significantly weaker supervision requirement. In the final version of the paper, we will make this supervision difference clearer and more explicit.

# 2 Additional experiments

We agree with Reviewer #3 that training curves will be a useful addition to our paper. We generated a set of curves to include in the final version, which show how the progress on non-elementary programs starts once a good performance has been reached with all the programs of lower level in the hierarchy. It can be observed that, for the BUBBLESORT program, the performance converges towards almost 100% of success after approximately 250 iterations, which corresponds to the training on $250 \times 20 = 5000$ traces. In the original NPI work, the network was trained on 1216 BUBBLESORT execution traces, which represents much more data as each trace contains also sub-traces corresponding to all the executed sub-programs, while in our setting traces are limited to the actions of a single program. Yet, our results exhibit better generalization properties. In future work, we will estimate more precisely the sample efficiency and generalization of AlphaNPI.

Regarding additional baselines, we will include the results of Cai et al. and Xiao et al.

# 3 Future work

The necessary verifiability of post-conditions is a current limitation of our work, as in some contexts post-conditions may not be easy to check. However, without post-conditions, new sub-programs would have to be defined during training, and the discovery of relevant sub-programs is a very difficult challenge. Removing the requirement of post-conditions while keeping the global program easily interpretable would probably be a very difficult extension to our work. Instead, in immediate future work we intend to remove the need for the hierarchy of program levels, resulting in the need for more exploration and a less immediate curriculum learning strategy. We intend to use curriculum learning based on learning progress (LP) to overcome the need for an explicit hierarchy given in advance. With this strategy, the agent should be able to focus its training on low-level programs which are easier to learn and thus provide high LP before attempting training on higher-level ones.

[Meta-Review · NeurIPS 2019]

The authors should be commended for an excellent submission to NeurIPS. The concerns about clarity the reviewers raised seem to be addressable as the authors describe in their rebuttal. The topic: "unsupervised" (really, less-supervised) structured neural program induction is perfect for NeurIPS and the empirical results on sorting and other tasks as compared to the original neural programmer interpreter are exciting.